# Inverse Association between Statin Use and Cancer Mortality Relates to Cholesterol Level

**DOI:** 10.3390/cancers14122920

**Published:** 2022-06-14

**Authors:** Antti I. Peltomaa, Kirsi Talala, Kimmo Taari, Teuvo L. J. Tammela, Anssi Auvinen, Teemu J. Murtola

**Affiliations:** 1Faculty of Medicine and Health Technology, Tampere University, 33100 Tampere, Finland; teuvo.tammela@tuni.fi (T.L.J.T.); teemu.murtola@tuni.fi (T.J.M.); 2Finnish Cancer Registry, 00100 Helsinki, Finland; kirsi.talala@cancer.fi; 3Department of Urology, University of Helsinki and Helsinki University Hospital, 00100 Helsinki, Finland; kimmo.taari@hus.fi; 4Department of Urology, TAYS Cancer Center, 33100 Tampere, Finland; 5Faculty of Social Sciences, Tampere University, 33100 Tampere, Finland; anssi.auvinen@tuni.fi

**Keywords:** statins, cancer, cholesterol

## Abstract

**Simple Summary:**

We observed that the inverse association between statin use and cancer mortality is limited to men with a reduction in cholesterol after the commencement of statins. These findings demonstrate that the observed inverse association between the use of statins and mortality from cancer is related to cholesterol level. To our knowledge, this is the first study to evaluate the separate effects of cholesterol level and statin use on cancer mortality.

**Abstract:**

Statins have been associated with a decreased cancer mortality. However, cholesterol level as such may modify the risk of cancer death. To clarify the complex interplay between statins, cholesterol level, and cancer mortality, we conducted a comprehensive analysis to separate the effects of cholesterol level and statin medication on cancer mortality. Our study population consisted of 16,924 men participating in the Finnish Randomized Study of Screening for Prostate Cancer with at least one cholesterol measurement during follow-up (1996–2017). Cox proportional regression was used to estimate hazard ratios. In total, 1699 cancer deaths were observed during the median follow-up of 19 years. When statins’ association with the risk of cancer death was estimated without adjustment for cholesterol level, statin use was associated with a lowered cancer mortality (HR 0.87; 95% CI 0.79–0.97) compared to non-users. However, with further adjustment for total cholesterol level, statin use was no longer associated with a lower cancer mortality (HR 1.08; 95% CI 0.97–1.20). Upon stratified analysis, statin use was associated with a decreased cancer mortality only if the total cholesterol level decreased after the initiation of statin use (HR 0.66; 95% CI 0.58–0.76). The inverse association between statin use and cancer mortality is limited to men with a reduction in total cholesterol level after the commencement of statins, i.e., statin use is associated with a lowered cancer mortality only if the total cholesterol level decreases. This suggests that the effect of statin use on cancer mortality relates to the decreased total cholesterol level.

## 1. Introduction

Cholesterol plays an important role in cellular membranes, energy metabolism, and signal mediation [1]. In addition to these essential functions in normal cells, tumorigenesis is crucially dependent on cholesterol metabolism [2]. The aberrant lipid metabolism in cancer cells has been a fascinating area of recent research [3,4]. Both the de novo synthesis and exogenous uptake of fatty acids by cancer cells have been reported as a means of satisfying the increased demand for nutrients required for the proliferation of cancer [4]. Cholesterol is also needed in the tumor microenvironment as a part of lipid rafts which have been shown to be a part of survival signaling and are increasingly expressed in cancer cells [5].

The association between cholesterol levels and cancer risk or cancer mortality has been studied thoroughly [6,7,8]. Ahn et al. suggested in their study that the detected association between a high cholesterol level and the risk of cancer might be explained by reverse causation [8]. A comprehensive review by Kuzu et al. in 2016 also pointed out the controversy in the results of previous epidemiological studies assessing the risk of cancer and cholesterol level [7]. However, they highlight many potential mechanisms by which medications (e.g., statins, squalene synthesis inhibitors, and farnesyl or geranylgeranyl inhibitors) affecting cholesterol levels might modify cancer metabolism. Cholesterol level and the risk of cancer death have been shown to obey a U-curve association. An especially low cholesterol level has been linked to an increased risk of cancer death in previous studies [9,10], but similarly to the results from studies assessing the association between cholesterol level and the risk of cancer, these studies are prone to reverse causation bias.

In addition, the effects of statin use on cancer mortality or cancer risk have been under intensive research [11,12]. The incidence of cancer has not been reliably associated with statin use [11], but there is increasing evidence that statins are associated with a lowered cancer mortality, especially in some cancer types [12].

Reverse causation poses a major challenge for studies evaluating the association between statin use and the risk of cancer death. The association between low cholesterol levels and an increased risk of cancer has been attributed mainly to reverse causation (i.e., undiagnosed cancer causes a reduction in cholesterol level), although a modestly increased long-term risk of cancer has been difficult to rule out [13]. Considering cholesterol level’s major role in the decision to prescribe statins, low cholesterol level due to undiagnosed cancer may also decrease the probability of initiation of statins, and therefore modify the association between statin use and the risk of cancer death.

However, few studies have been able to assess statins’ effect on cancer mortality in conjunction with cholesterol levels. To clarify this complex interplay between cholesterol levels, statins, and cancer mortality, we assessed the risk of cancer death considering both statin use and cholesterol levels in a comprehensive analysis of men participating in the Finnish Randomized Study of Screening for Prostate Cancer (FinRSPC). Our hypothesis is that the inverse association between statin use and cancer mortality is mediated by the underlying cholesterol level.

## 2. Materials and Methods

### 2.1. Study Cohort

The FinRSPC is a randomized population-based trial assessing the effect of systematic screening with prostate-specific antigen (PSA) on prostate cancer mortality. The FinRSPC study protocol (approved by the Ethics Port of Pirkanmaa Hospital; decision number ETL95077) has been described comprehensively previously [14]. We identified all the men from the Tampere metropolitan region free of cancer at baseline with at least one cholesterol measurement during the follow-up (1996–2017), and those 16,924 men formed the study cohort for the present study. Follow-up started at FinRSPC randomization in 1996–1999 and continued until death, emigration from Finland, or 31 December 2017, whichever occurred first.

Statistics Finland registers all deaths occurring in Finland. Cancer deaths are defined using ICD-10 codes C00–D48 recorded as a primary cause of death. In analyses, where deaths by specific cancer type were used, the ICD-10 codes were defined as follows: lung cancer C34, colorectal cancer C18–C20, and pancreatic cancer C25. Statistics Finland gave permission to use the cause of death data (TK-53-1330-18).

### 2.2. Information on Medication Use and Cholesterol Values

The study cohort was linked to two databases: the National Prescription Database maintained by the Social Insurance Institute of Finland (SII) and the Fimlab laboratory service database. Fimlab registers the majority of laboratory results in the Pirkanmaa region and is the primary distributor of laboratory services in the Pirkanmaa district. Information on total cholesterol measurements (*n* = 16,924), LDL (*n* = 15,425), HDL (*n* = 15,625), and triglyceride (*n* = 17,043) measurements was acquired, and yearly mean values for each person were calculated for each follow-up year. Men with lipid measurements available for a given year were stratified into two groups based on the following threshold values: total cholesterol level of 5.0 mmol/L, HDL level of 1.0 mmol/L, LDL level of 3.0 mmol/L, and triglyceride level of 1.7 mmol/L. If no lipid measurements were available for a given year, those men were categorized into a separate missing category.

The SII provides reimbursements for purchases of physician-prescribed medications in Finland and registers all purchases in a way that has been previously described in detail [15]. Information on all prescribed statin, antidiabetic, antihypertensive, and non-steroidal anti-inflammatory (NSAID) drug purchases during follow-up was obtained. Each year with recorded reimbursed purchases was considered as a year of usage. For statins, we calculated the mg amount of statin purchases during a given calendar year and standardized different statins by dividing the total mg amount of statins by the amount corresponding to a defined daily dose (DDD) as defined by the World Health Organization [16]. Cumulative statin DDDs and years of usage were calculated by adding together the total statin DDDs or years of usage, respectively. The statin-use intensity (average DDDs/year) was calculated by dividing the total DDDs by the cumulative years of statin usage.

We investigated the role of cholesterol-level change during statin use in a subgroup of men with data on cholesterol measurements available both before and after the initiation of statin use. A total of 3710 men formed this subgroup, and there were 335 cancer deaths during the median follow-up of 19.1 years. We quantified the change in cholesterol by subtracting the first cholesterol measurement after the initiation of statin use from the latest measurement before it. Statin users were then divided by total cholesterol, LDL, or triglyceride level modification stratified as dichotomous (no decrease, any decrease) or a trichotomous variable (no decrease, decrease at median or below, decrease above median; i.e., median decrease was 1.53 mmol/L for total cholesterol and it was used as a cut-point for the trichotomous variable).

The trial population was linked to the Care Register for Health Care (HILMO) maintained by the National Institute for Health and Welfare to obtain diagnoses from inpatient periods during the follow-up. We calculated a modified Charlson comorbidity index by utilizing hospital episode diagnoses until the year 2000. Previously, the Charlson comorbidity index based on hospital episode statistics has been shown to predict mortality [17].

### 2.3. Statistical Analysis

Cox proportional hazards regression was used to estimate the hazard ratios (HR) and 95% confidence intervals (95% CI) for overall cancer death, and separately for three commonly fatal cancer types (lung cancer, colorectal cancer, and pancreatic cancer). We used adjustments for age at randomization, FinRSPC trial arm, Charlson comorbidity index, and simultaneous use of antidiabetic, antihypertensive, NSAIDs, aspirin, and anticoagulants. Simultaneous drug use was analyzed as a time-dependent variable and all other adjustments as fixed (time-constant) variables. We also assessed cardiovascular mortality by conducting a competing-risks analysis.

Drug-use status was allowed to change on a yearly basis; for statins, the cumulative amount, duration, and intensity were also updated for each follow-up year based on the recorded drug purchases. After the first drug reimbursement, the status remained as a user even when purchases stopped in order to limit bias due to the selective discontinuation of statins in palliative care. The drug-use variables were time-dependent in all analyses. Dichotomous cholesterol variables were also allowed to change on a yearly basis for each follow-up year. The statistical significance of interaction was assessed by adding an interaction term to the Cox regression model.

We estimated latency of risk associations by lagging statin or cholesterol exposure for a fixed amount of years, i.e., in a 3-year lag-time analyses we used statin or cholesterol statuses that had occurred 3 years earlier.

## 3. Results

### 3.1. Population Characteristics

Of the 16,924 FinRSPC participants who had data on cholesterol measurements available during the follow-up, 9555 (56.4%) had used statins (Table 1). During the median follow-up of 19.9 and 18.9 years, there were 791 and 908 cancer deaths among statin users and non-users, respectively. In comparison, there were 1050 and 649 deaths among patients with a mean total cholesterol level below 5 mmol/L and above 5 mmol/L, respectively. Both statin users and patients with lower cholesterol values were more likely to use co-medications, i.e., NSAIDs, aspirin, antihypertensive, and antidiabetic drugs.

### 3.2. Risk of Cancer Death by Serum Cholesterol Level

The dichotomous mean serum total cholesterol level during follow-up was not associated with the risk of cancer death (HR 0.84 95% CI 0.65–1.08) (Table 2). However, an LDL above 3 mmol/L and an HDL above 1 mmol/L were associated with a slightly decreased risk of cancer death (HR 0.71; 95% CI 0.53–0.95 for LDL above 3 mmol/L, and HR 0.53; 95% CI 0.42–0.67 for HDL above 1 mmol/L). This risk decrease remained during lag time analysis when the exposure was lagged 1 year for LDL and up to 3 years for HDL. The mean triglyceride level was not associated with the risk of cancer death during the main analysis or lag-time analyses.

### 3.3. Risk of Cancer Death by Statin Use after Randomization in the FinRSPC

When assessing the effects of statins without taking into account the cholesterol level, the risk of overall cancer death was slightly decreased among statin users (HR 0.87; 95% CI 0.79–0.97) (Table 3). A statistically significant risk decrease was also observed in the risk of colorectal cancer death (HR 0.58 95% CI 0.41–0.80). High-intensity statin users (tertile with the highest statin use amount/usage year; >219 DDD/year) had a decreased risk of colorectal, pancreatic, and overall cancer death. Similar decreasing risk trends for the duration and amount of statin use were observed.

We performed a sensitivity analysis with further adjustment for marital and employment status. The results remained similar to the main analysis (HR 0.88; 95% CI 0.79–0.97; risk of overall cancer death by statin use).

### 3.4. Risk of Cancer Death by Statin Use and Cholesterol Level after Randomization in the FinRSPC

Statin use was not associated with a decreased risk of overall cancer death when adjustment for total cholesterol level was used as a dichotomous time-dependent variable (HR 1.08; 95% CI 0.97–1.20) (Table 4). No risk decrease was observed for lung, colorectal, pancreatic, or other cancer types either (the results for other cancer types are presented in Appendix A). In a lag-time analysis adjusted for the total cholesterol level, the results were similar and no effect modification was observed for the overall, lung, colorectal or pancreatic cancer death when statin use was lagged by 1, 3, or 5 years from the initial timing of statin exposure. The total cholesterol level was not associated with an increased or decreased risk of cancer death in this model either. In a sensitivity analysis in which cardiovascular death was included as a competing risk, statin use was associated with a slightly increased risk of death (HR 1.54; 95% CI 1.38–1.71).

### 3.5. Cancer Mortality in Relation to Change in Cholesterol Level after Initiation of Statin Use

Statin use was not associated with a decreased risk of cancer death among participants whose total cholesterol level did not decrease after the initiation of statin use (HR 0.97; 95% CI 0.62–1.51). Instead, statin use was associated with a lowered cancer mortality compared to non-users when the initiation of statin use led to a decrease in total cholesterol level (HR 0.65; 95% CI 0.54–0.78 and HR 0.61; 95% CI 0.51–0.73 for a decrease in the total cholesterol level of 1.53 mmol/L or lower and 1.53 mmol/L or higher, respectively). However, the effect modification by cholesterol change after the initiation of statin use was not statistically significant (P for interaction 0.09). A similar risk modification was not observed for the lipoprotein subtypes LDL and triglycerides (Table 5).

## 4. Discussion

Our results show that the reduced risk of cancer death among statin users is limited to men with a reduction in serum total cholesterol following the initiation of statin use. This finding may explain previous conflicting results on statins’ effects on cancer mortality [13,18,19,20,21,22]. In our study, we analyzed the overall cancer mortality and three hormone-independent cancers (colorectal, lung, and pancreatic cancer) which are common causes of cancer death, and the results were similar in all cancer types.

As far as we know, this is the first study to assess the time-dependent effects of both cholesterol level and statin use on cancer mortality. We were able to include cholesterol level in our analysis as a time-dependent dichotomous variable to evaluate statins’ and cholesterol level’s individual effects on the risk of cancer death more completely.

In a recent meta-analysis [23] that pooled 60 studies assessing statins’ effects on cancer mortality, statin use was associated with a slightly decreased cancer mortality and recurrence-free survival. Similarly, we found a decreased cancer mortality among statin users, but that was limited to men whose cholesterol level decreased after the initiation of statin use. This suggests that an inverse association between statin use and cancer mortality may be linked to a reduction in cholesterol levels. Previously, cholesterol levels have been shown to decrease in advanced cancer [11], and therefore cancer patients or even patients with occult cancer may be less likely to initiate cholesterol-lowering medications, which may have caused bias in the results of previous epidemiological studies.

In this study, total cholesterol level was not associated with the risk of cancer death, but surprisingly, both higher HDL and LDL were associated with a slightly decreased risk of cancer death. The practice in cholesterol measurements has changed during the follow-up period; there were more patients with total cholesterol measurements than HDL/LDL measurements, and that difference might explain a slight discrepancy in the hazard ratios. On the other hand, it has been suggested that certain lipoproteins such as HDL could be especially important for prostate cancer cells [24]. For practical reasons, we concentrated in this analysis on serum cholesterol, and intracellular cholesterol was beyond the scope of this study. In a mouse model of prostate cancer, serum cholesterol has been shown to decrease intraprostatic androgens and tumor growth, suggesting that serum cholesterol acts as a plausible surrogate for the intracellular cholesterol level [25].

The role of cholesterol may be different in hormone-dependent and hormone-independent cancer types. In hormone-independent cancer types, cholesterol is needed for cellular membranes and cancer growth. Additionally, cholesterol is a precursor for steroid hormone synthesis, which plays a major role—especially in hormone-dependent cancer types. This difference in cholesterol metabolism may explain the observation that the previously reported inverse association between statin use and cancer mortality seem to be the most consistent in hormone-dependent cancer types (ovarian, breast, endocrine-related gynecological cancer, and prostate cancer) [13,26,27,28,29,30,31]. Our results are consistent with the notion that, at least in hormone-independent cancer types, statins have no independent effect on cancer mortality, but the mortality reduction is linked to the ensuing decrease in total cholesterol level. Our subgroup analysis, stratified by the change in total cholesterol level after initiation of statin use, supported this notion. As we have information only on drug purchases, we cannot know for certain whether participants had taken their statins as prescribed, but a change in cholesterol level can be presumed as a marker of treatment compliance. Our findings suggest that a decrease in the total cholesterol level, instead of statin use per se, is behind the association between statin use and reduced cancer mortality. Thus, cholesterol level is likely more important than statin use.

The suppression of isoprenoid production, another consequence of statin-induced mevalonate pathway inhibition, has also been shown to affect carcinogenesis [32]. In vitro models have proposed statins’ anticancer effects to be partly mediated by isoprenoid depletion [33]. However, some isoprenoids (e.g., geranylgeraniol, geranylgeranoic acid, and tocotrienols) seem to have either independent or synergistic inhibitory effects with statins on the proliferation of cancer cells in vitro [34,35,36]. More information is still needed to conclude the net effect of the statin-induced decrease in isoprenoid production in cancer patients, but it is already known that isoprenoids have the potential to act as an adjuvant agent in reducing statin-induced toxicities in cancer therapy [34]. In our present study, statin use did not have an independent risk association with cancer mortality after adjustment for cholesterol level, which suggests that at the population level, the inhibition of cholesterol production may be more important than the inhibition of isoprenoid production. In addition, our previous work found that at low statin doses, such as those seen in standard oral dosing, only cholesterol reduction may be affected. The inhibition of isoprenoid production might require higher statin concentrations than those achieved in the serum during standard clinical dosing [37].

This area of research is both intriguing and challenging, with many potential pitfalls and possible sources of bias regarding statin use in cancer patients [38]. The main indication for the use of statins is the prevention of cardiovascular events, and usually, the need for statins is assessed individually depending on the patient’s risk profile [39]. In cancer patients with a limited life-expectancy, medications such as statins are usually discontinued when the benefit of drug use is assessed to be lower than its potential harms, and hence criteria for initiating statins differ compared to patients without cancer [40]. To manage the bias caused by the tendency to discontinue statins in late-stage terminal cancer, we ignored the discontinuation of statin use during the follow-up. Statin users are also known to differ from non-users by socioeconomic status and co-medication use [41]. To minimize potential healthy user bias, we used a time-dependent adjustment for co-medication use (NSAIDs, aspirin, antihypertensives, antidiabetics, and anticoagulants) and the modified Charlson comorbidity index. Immortal time bias was addressed by using time-dependent variables with exposure beginning from the first purchase of a prescribed statin medication.

We were able to analyze the effect of statins and cholesterol values in our large population-based cohort, which consisted of 16,924 men with at least one cholesterol measurement during the follow-up in the FinRSPC trial. We had access to reliable and comprehensive national databases on medication use, laboratory data, and causes of death, allowing us to conduct a multi-variable analysis with time-dependent adjustment for other medications and cholesterol values. However, we did not have information on behavioral factors such as smoking, physical activity, diet, cancer stage at diagnosis or the use of health services; therefore, residual confounding cannot be excluded. In addition, treatment choices between patients may differ by statin use status, and this may also be an additional source of confounding. Our study shows that the inverse association between statin use and cancer mortality likely relates to the ensuing decrease in total cholesterol level, suggesting that total cholesterol level may be an important underlying factor in the lowered cancer mortality among statin users.

## 5. Conclusions

In our well-characterized cohort, we used detailed information on cholesterol and statin use and found a slight inverse association between cholesterol level and the risk of cancer death. The previously reported inverse association between statin use and cancer mortality was observed among statin users with reduced total cholesterol levels, while no such effect was seen in users without a total cholesterol reduction. Our results show that the effect of statins on the risk of cancer death relates to the change in total cholesterol level.

## Figures and Tables

**Table 1 cancers-14-02920-t001:** Baseline characteristics of study population. Cohort of participants with at least one serum total cholesterol available in the Finnish Randomized Study of Screening for Prostate Cancer between 1996 and 2017. NA: not acceptable.

				Mean Total Cholesterol (mmol/L) during Follow-Up	
	Cholesterol Cohort	Any Statin Use	No Statin Use	5 or Below	Above 5
Participants (*n*)	16,924	9555	7369	10,301	6623
Median (IQR) age at baseline (years)	59 (55–63)	59 (55–63)	59 (55–63)	59 (55–63)	59 (55–63)
Number of deaths	6316	3157	3159	3899	2417
Median (IQR) follow-up time (years)	19.0 (16.4–20.9)	19.9 (18.0–20.9)	18.9 (14.3–20.9)	19.3 (16.8–20.9)	18.9 (15.7–20.9)
Median (IQR) body mass index (kg/m^2^)	26.8 (24.7–29.5)	27.2 (25.1–29.9)	26.3 (24.3–29.1)	27.1 (24.9–29.9)	26.4 (24.3–29.1)
Median (IQR) amount of cholesterol measurements	5 (2–8)	7 (4–10)	3 (1–5)	6 (3–9)	4 (2–7)
Marital status					
Married/registered partnership	13,072 (77.2%)	7632 (79.9%)	5440 (73.8%)	7959 (77.3%)	5113 (77.2%)
Not married	3693 (21.8%)	1827 (19.1%)	1866 (25.3%)	2242 (21.8%)	1451 (21.9%)
Divorced	94 (0.6%)	58 (0.6%)	36 (0.5%)	61 (0.6%)	33 (0.5%)
Widow	65 (0.4%)	38 (0.4%)	27 (0.4%)	39 (0.4%)	26 (0.4%)
Employment status					
Employed	7518 (44.4%)	4327 (45.3%)	3191(43.3%)	4433 (43.0%)	3085 (46.6%)
Unemployed	2053 (12.1%)	1086 (11.4%)	967 (13.1%)	1230 (11.9%)	823 (12.4%)
Retired	7181 (42.4%)	4051 (42.4%)	3130 (42.5%)	4540 (44.1%)	2641 (39.9%)
Drug usage					
Use of statin drugs; *n* (%)	9555 (56.5%)	NA	NA	6306 (61.2%)	3249 (49.1%)
Use of antihypertensive drugs; *n* (%)	13,973 (82.5%)	8729 (92.5%)	5244 (71.2%)	8973 (87.1%)	5000 (75.5%)
Use of antidiabetic drugs; *n* (%)	3874 (22.9%)	2992 (31.7%)	882 (12.0%)	2975 (28.9%)	899 (13.6%)
Use of aspirin; *n* (%)	3695 (21.8%)	2705 (28.7%)	990 (13.4%)	2506 (24.3%)	1189 (18.0%)
Use of NSAID drugs; *n* (%)	14,944 (88.3%)	8636 (91.5%)	6308 (85.6%)	9186 (89.2%)	5758 (86.9%)
Cause of death					
All cancers	1699	791	908	1050	649
Lung cancer	424	205	219	259	165
Colorectal cancer	168	65	103	104	64
Pancreatic cancer	139	61	78	91	48
Gastric cancer	67	28	39	38	29
Liver cancer	83	34	49	64	19
Non-Hodgkin lymphoma	58	33	25	40	18
Kidney cancer	49	31	18	31	18
Bladder cancer	48	24	24	31	17
Brain and CNS cancers	44	21	23	27	17

**Table 2 cancers-14-02920-t002:** Risk of overall cancer death by serum cholesterol level and lipid fractions in a cohort of the Finnish Randomized Study of Screening for Prostate Cancer patients.

		Main Analysis	Lag-Time Analysis		
			1 Year	3 Years	5 Years
	N of Deaths	HR (95% CI)	HR (95% CI)	HR (95% CI)	HR (95% CI)
Serum cholesterol (mmol/L)					
5 or lower	158	Ref	Ref	Ref	Ref
above 5	131	0.84 (0.65–1.08)	0.88 (0.73–1.05)	0.90 (0.76–1.05)	0.88 (0.74–1.04)
LDL cholesterol (mmol/L)					
3 or lower	162	Ref	Ref	Ref	Ref
above 3	83	0.71 (0.53–0.95)	0.80 (0.66–0.98)	0.90 (0.75–1.07)	0.88 (0.72–1.07)
Triglycerides (mmol/L)					
1.7 or lower	215	Ref	Ref	Ref	Ref
above 1.7	77	0.93 (0.72–1.19)	1.00 (0.83–1.20)	0.99 (0.84–1.18)	0.99 (0.83–1.19)
HDL cholesterol (mmol/L)					
1.0 or lower	47	Ref	Ref	Ref	Ref
above 1.0	202	0.53 (0.42–0.67)	0.77 (0.64–0.93)	0.80 (0.67–0.97)	1.07 (0.86–1.33)

**Table 3 cancers-14-02920-t003:** Risk of cancer death by amount, duration, and intensity of statin use in the Finnish Randomized Study of Screening for Prostate Cancer.

	All Cancers		Lung Cancer		Colorectal Cancer		Pancreatic Cancer	
	N of Deaths *	HR (95% CI) **	N of Deaths	HR (95% CI)	N of Deaths	HR (95% CI)	N of Deaths	HR (95% CI)
Non-users	907	Ref	219	Ref	103	Ref	78	Ref
Any users	792	0.87 (0.79–0.97)	205	1.07 (0.87–1.32)	65	0.58 (0.41–0.80)	61	0.70 (0.49–1.01)
Amount of statin use (DDDs)								
<1024 DDD	364	0.98 (0.79–1.22)	100	1.10 (0.72–1.70)	24	0.56 (0.25–1.28)	25	0.74 (0.32–1.69)
1024–2691 DDD	276	0.93 (0.83–1.04)	70	1.16 (0.93–1.45)	25	0.57 (0.39–0.83)	24	0.79 (0.53–1.16)
>2691 DDD	152	0.63 (0.52–0.77)	35	0.71 (0.47–1.07)	16	0.53 (0.29–0.96)	12	0.45 (0.22–0.92)
Duration of statin use (years)								
<6 years	401	0.93 (0.84–1.04)	116	1.16 (0.94–1.43)	27	0.55 (0.38–0.80)	24	0.83 (0.57–1.20)
6–12 years	253	0.66 (0.54–0.80)	57	0.77 (0.51–1.16)	22	0.58 (0.32–1.05)	27	0.39 (0.18–0.85)
>12 years	138	0.71 (0.51–0.98)	32	0.67 (0.31–1.44)	16	0.57 (0.21–1.60)	10	0.36 (0.09–1.49)
Intensity of statin use (DDDs/year)								
<103	443	0.99 (0.82–1.19)	120	1.02 (0.69–1.50)	33	0.52 (0.25–1.07)	37	0.94 (0.49–1.77)
103–219	233	0.99 (0.87–1.12)	52	1.35 (1.06–1.71)	22	0.59 (0.39–0.91)	16	0.76 (0.48–1.19)
>219	116	0.69 (0.60–0.80)	33	0.77 (0.57–1.05)	10	0.54 (0.34–0.86)	8	0.56 (0.33–0.94)

* Number of deaths among all statin users (users and previous users) and non-users. ** Calculated using Cox regression with adjustment for age and use of other medications: aspirin and other NSAIDs, antihypertensive and antidiabetic drugs, alpha-blockers, and 5α-reductase inhibitor.

**Table 4 cancers-14-02920-t004:** Overall cancer mortality by statin use and total cholesterol level (mmol/L) in the Finnish Randomized Study of Screening for Prostate Cancer.

	Main Analysis	Lag-Time Analysis *		
		1 Year	3 Years	5 Years
	HR (95% CI)	HR (95% CI)	HR (95% CI)	HR (95% CI)
All cancers				
Non-users	Ref	Ref	Ref	Ref
Statin users	1.08 (0.97–1.20)	0.93 (0.81–1.06)	0.91 (0.81–1.03)	0.90 (0.80–1.02)
Total cholesterol below 5	Ref			
Total cholesterol above 5	0.85 (0.66–1.10)			
Lung cancer				
Non-users	Ref	Ref	Ref	Ref
Statin users	1.35 (1.10–1.67)	1.09 (0.85–1.41)	0.98 (0.77–1.25)	1.02 (0.81–1.30)
Total cholesterol below 5	Ref			
Total cholesterol above 5	0.81 (0.49–1.36)			
Colorectal cancer				
Non-users	Ref	Ref	Ref	Ref
Statin users	0.73 (0.48–1.24)	0.79 (0.51–1.22)	0.80 (0.54–1.18)	0.64 (0.43–0.96)
Total cholesterol below 5	Ref			
Total cholesterol above 5	0.82 (0.33–2.05)			
Pancreatic cancer				
Non-users	Ref	Ref	Ref	Ref
Statin users	0.88 (0.61–1.28)	0.68 (0.42–1.12)	0.71 (0.45–1.11)	0.74 (0.48–1.14)
Total cholesterol below 5	Ref			
Total cholesterol above 5	1.35 (0.61–2.99)			

* Statin exposure was lagged 1, 3, or 5 years, i.e., in a 3-year lag-analysis we used the statin-use status occurring 3 years earlier.

**Table 5 cancers-14-02920-t005:** Overall cancer mortality by statin use stratified by change in cholesterol level after initiating statin medication.

	N of Men	N of Deaths	HR (95% CI)
Statin Non-Users			Ref
Statin users’ change in total cholesterol after statin initiation			
No change or increase	170	87	0.97 (0.62–1.51)
Decrease < 1.53 mmol/L *	1680	629	0.65 (0.54–0.78)
Decrease > 1.53 mmol/L	1858	768	0.61 (0.51–0.73)
Any decrease	3538	1397	0.66 (0.58–0.76)
P for interaction by cholesterol change **			0.09
Statin users’ change in LDL level after statin initiation			
No change or increase	162	50	0.51 (0.27–0.96)
Decrease < 0.96 mmol/L *	433	135	0.52 (0.36–0.75)
Decrease > 0.96 mmol/L	604	204	0.69 (0.52–0.91)
Any decrease	1037	339	0.62 (0.50–0.78)
P for interaction by LDL change **			0.63
Statin users’ change in triglyceride level after statin initiation			
No change or increase	990	423	0.65 (0.52–0.81)
Decrease < 0.30 mmol/L *	926	345	0.66 (0.52–0.84)
Decrease > 0.30 mmol/L	1888	747	0.61 (0.51–0.73)
Any decrease	2814	1092	0.65 (0.56–0.76)
P for interaction by triglyceride change **			0.76

* Cut-points represent the median decrease in total cholesterol, LDL, and triglyceride level, respectively. ** Calculated for the dichotomous cholesterol change variable.

## Data Availability

The data that support the findings of this study are available on request from the corresponding author. The data are not publicly available due to their containing information that could compromise the privacy of research participants.

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
