# Peer review of "Inverse Association between Statin Use and Cancer Mortality Relates to Cholesterol Level"

_cancers, 2022, doi:10.3390/cancers14122920_

Round 1

Reviewer 1 Report

This manuscript establishes the effect of statins on survival after carcinogenesis under limited conditions. The manuscript was well prepared. Moreover, the cited and their discussion are presented in good style. However, authors have a point that needs to be addressed.

The use of statins that block the upstream of the MVA pathway not only inhibits cholesterol synthesis, but also biosynthesis of the isoprenoids. Therefore, when examining the association of statins with carcinogenesis inhibition or carcinogenesis, examination of cholesterol alone may obscure conclusions. For example, cancer-preventive isoprenoid such as geranylgeraniol (PMID: 4352794, 11572758) and geranylgeranoic acid (PMID: 34564450, 30622150). Therefore, in this manuscript, it is necessary to mention the carcinogenic inhibitory effect of isoprenoids and comprehensively discuss the benefits of cholesterol reduction and the disadvantages of isoprenoid depletion by statin. Especially, for the potential mechanisms of "L50-L52 However, they highlight many potential mechanisms by which medications (e.g. statins, squalene synthesis inhibitors and farnesyl or geranylgeranyl inhibitors) affecting cholesterol levels might modify cancer metabolism.", should cite them as previous studies have reported isoprenoids that prevent cancer or, conversely, aggravate cancer.

Author Response

COMMENT: “The use of statins that block the upstream of the MVA pathway not only inhibits cholesterol synthesis, but also biosynthesis of the isoprenoids. Therefore, when examining the association of statins with carcinogenesis inhibition or carcinogenesis, examination of cholesterol alone may obscure conclusions. For example, cancer-preventive isoprenoid such as geranylgeraniol (PMID: 4352794, 11572758) and geranylgeranoic acid (PMID: 34564450, 30622150). Therefore, in this manuscript, it is necessary to mention the carcinogenic inhibitory effect of isoprenoids and comprehensively discuss the benefits of cholesterol reduction and the disadvantages of isoprenoid depletion by statin. Especially, for the potential mechanisms of "L50-L52 However, they highlight many potential mechanisms by which medications (e.g. statins, squalene synthesis inhibitors and farnesyl or geranylgeranyl inhibitors) affecting cholesterol levels might modify cancer metabolism.", should cite them as previous studies have reported isoprenoids that prevent cancer or, conversely, aggravate cancer.”

RESPONSE: We totally agree with the importance of isoprenoids in this context. We have now added in the Discussion section a paragraph speculating the potential role of isoprenoids:

“The suppression of isoprenoid production, another consequence of statin-induced mevalonate pathway inhibition, has also been shown to affect carcinogenesis [32]. In vitro models have proposed statins’ anticancer effects to be partly mediated by isoprenoid depletion [33]. However, some isoprenoids (e.g., geranylgeraniol, geranylgeranoic acid and tocotrienols) seem to have either independent or synergistic inhibitory effect with statins on proliferation of cancer cells in vitro [34-36]. More information is still needed to conclude what is the net effect of statin-induced decrease in isoprenoid production in cancer patients, but it is already known that isoprenoids have potential to act as an adjuvant agent in reducing statin-induced toxicities in cancer therapy [34]. In our present study, statin use did not have independent risk association with cancer mortality after adjustment for cholesterol level, which suggests that at the population level inhibition of cholesterol production may be more important than inhibition of isoprenoid production. In addition, our previous work found that on low statin doses, such as those seen in standard oral dosing, only cholesterol reduction may be affected. Inhibition of isoprenoid production might require higher statin concentrations than those achieved in the serum during standard clinical dosing [37].”

Especially, the results showing synergistic effect of statins and isoprenoids are interesting and the hypothesis deserves more evaluation in the future.

Author Response

COMMENT: “While the authors have stated that “The FinRSPC study protocol has been described
comprehensively previously [14], it is important to mention the human IRB protocol
approval number associated with this study.”

RESPONSE: We have now added approval number in Materials and Methods section Ethics Port of Pirkanmaa Hospital has approved the FinRSPC study protocol, decision number ETL95077”.

Reviewer 3 Report

Dear Authors,

The article on the complex interplay between statins, cholesterol level and cancer mortality found a slight inverse association between cholesterol level and risk of cancer death that relates to change in total cholesterol level. This is important for practice.

Comments:

- it would be useful to have the units of measures (mmol/l) for mean total cholesterol level below 5, similar to HDL-Col.

- table 1: it is not clear the text explanation "cholesterol measurements available during the follow-up, 9,555 (56.4%)" with the table information "Use of statin drugs; n (%)  9,438 (55.7 %)    NA   NA   6,261 (60.8%)   3,117 (47.0%)".

Author Response

COMMENT: “it would be useful to have the units of measures (mmol/l) for mean total cholesterol level below 5, similar to HDL-Col.”

RESPONSE: We have now revised the manuscript and tables. Missing units of measures were added.

COMMENT: table 1: it is not clear the text explanation "cholesterol measurements available during the follow-up, 9,555 (56.4%)" with the table information "Use of statin drugs; n (%)  9,438 (55.7 %)    NA   NA   6,261 (60.8%)   3,117 (47.0%)".

RESPONSE: Thank you for your notion. The information on Table 1 was only partly updated when one more follow-up year was added. There was a total of 9,555 statin users during the follow-up and of those 6,306 and 3,249 had mean total cholesterol below or above 5 mmol/l, respectively.

Round 2

Reviewer 3 Report

Dear Authors,

This article on the complex interplay between statins, cholesterol level and cancer mortality has practical importance through the slight inverse association between cholesterol level and risk of cancer death that relates to change in total cholesterol level. 

This manuscript is a resubmission of an earlier submission. The following is a list of the peer review reports and author responses from that submission.

Round 1

Reviewer 1 Report

This manuscript establishes the effect of statins on survival after carcinogenesis under limited conditions. The manuscript was well prepared. Moreover, the cited and their discussion are presented in good style. However, authors have a point that needs to be addressed.

Major point

In addition to what the authors discussed, the use of statins depletes cholesterol as a cell membrane component in the production of new cells. In addition, the biosynthesis of non-steroidal lipids, including geranylgeranyl diphosphate and farnesyl diphosphate, involved in prenylation of Rho proteins is also inhibited. Therefore, it is quite reliable that statins can be a means of combating cancer.

  1. Is there any possible reason other than hormone dependence and the origin tissues for cancer that were not effective in improving survival by statins?
  2. Is there any information other than Table 1 about the liver, which is the main organ of cholesterol synthesis?
  3. Contrary to previous findings, there are recent reports from the Journal of lipid research and Metabolites that geranylgeranoic acid, which is synthesized from GGPP via geranylgeraniol, suppresses carcinogenesis and statins inhibits this effect. Can these reports contribute to the discussion of the population in which statins were ineffective?

    Please let me and your readers know if you have any additional information.

Minor point

  1. Correct the line break error in L102-104.

  2. Check again about the style of the reference.

  3. Delete L20- (FinRSCP). The abbreviation is required only if it appears multiple times.

  4. On the other hand, since L55- (FinRSPC) is written in L52, there is no problem using the abbreviation from the beginning.

Reviewer 2 Report

See attachment.

Round 2

Reviewer 1 Report

The manuscript has been improved as requested.

For the future, if authors like, please also read the literature on PMID: 34564450, 30622150. 

The use of statins that block the upstream of the MVA pathway not only inhibits cholesterol synthesis, but also biosynthesis of the carcinogenic suppressive isoprenoids such as geranalgeranoic acid, and the results may not be clear at this time.

Reviewer 2 Report

See attachment.